# A Model between Cohesion and Its Inter-Controlled Factors of Fine-Grained Sediments in Beichuan Debris Flow, Sichuan Province, China

Qinjun Wang [1,2,3,4,*], Jingjing Xie [2,3], Jingyi Yang [2,3], Peng Liu [2,3], Dingkun Chang [2,3] and Wentao Xu [2,3]

1 International Research Center of Big Data for Sustainable Development Goals, Beijing 100094, China
2 Key Laboratory of Digital Earth Science, Aerospace Information Research Institute, Chinese Academy of Sciences, Beijing 100094, China
3 Yanqihu Campus, University of Chinese Academy of Sciences, Beijing 101408, China
4 Key Laboratory of the Earth Observation of Hainan Province, Hainan Aerospace Information Research Institute, Sanya 572029, China
* Correspondence: wangqj@radi.ac.cn

**Abstract:** Cohesion is the attraction between adjacent particles within the same material, which is the main inter-controlled factor of fine-grained sediment stability, and thus plays an important role in debris flow hazard early warning. However, there is no quantitative model of cohesion and its inter-controlled factors, including effective internal friction angle, permeability coefficient and density. Therefore, establishing a quantitative model of cohesion and its inter-controlled factors is of considerable significance in debris flow hazard early warning. Taking Beichuan county in southwestern China as the study area, we carried out a series of experiments on cohesion and its inter-controlled factors. Using the value of cohesion as the dependent variable and values of normalized density, normalized logarithm of permeability coefficient and normalized effective internal friction angle as the independent variables, we established a quantitative model of cohesion and its inter-controlled factors by the least-squares multivariate statistical method. Fitting of the model showed that its determination coefficient ($R^2$) was 0.61, indicating that the corresponding correlation coefficient (R) was 0.78. Furthermore, t-tests of the model showed that except for the *p* value of density, which was 0.05, those of other factors were less than 0.01, indicating that cohesion was significantly correlated to its inter-controlled factors, providing a scientific basis for debris flow hazard early warning.

**Keywords:** debris flow; fine sediments; cohesion; Beichuan





## 1. Introduction

Debris flow is gravity sediment flow with a large amount of soils and stones caused by rainstorms or snow/ice melting. It often causes houses to collapse and results in damaged roads, electricity lines and other facilities, thus posing a serious threat to the safety of local people's lives and property [1]. For example, on August 20, 2019, catastrophic debris flow in Sichuan affected 446,000 people, leading to a direct economic loss of 15.89 billion yuan [2]. Therefore, rapid debris flow early warning plays an important role in ensuring the safety of mountainous people.

There are numerical debris flow hazard simulation methods, which are essential for the development of a hazard early warning system [3–6], such as the full three-dimensional (3D) smoothed particle hydrodynamics (SPH) method, the modified MPS method, the coupled moving particle simulation–finite element method and the liquid–gas-like phase transition model in sand flow under microgravity. In such models, quantitative debris flow parameters are important to the debris flow hazard simulations.

With particle size (represented by diameter) less than 2 mm, fine-grained sediments are Quaternary sediments and the main materials that flow in water during debris flow.

Their stability is closely related to the debris flow initial water volume [7–15]. Therefore, debris flow early warning needs to quickly detect their stability. It is mainly controlled by external and internal factors. External factors include water sources, such as rainfall, rainfall intensity, runoff and topographic conditions, e.g., slope, surface coverage and structure. Internal factors include cohesion, permeability coefficient and effective internal friction angle [16–18].

Cohesion is the attraction within a material, such as electrostatic attraction, van der Waals force, cementation and valence bonds. In the case of effective stress, cohesion is obtained by reducing the friction from the total shear strength, which is the inter-controlled factor of the debris flow stability. Its value mainly reflects the strain capacity of soil to resist external stress, and is related to the effective internal friction angle (the internal friction between soil particles, mainly including the surface friction of soil particles and the bonding force between them), the permeability coefficient (the unit flow under the unit hydraulic gradient, indicating the difficulty of fluid passing through the pore skeleton), density (mass per unit volume) and moisture (the ratio of the weight of water contained in the soil to the weight of dry soil) [19–27].

As there is no quantitative model of cohesion and its inter-controlled factors in debris flow, carrying out cohesion research and discovering its inter-controlled factors to establish a quantitative model is of great significance in debris flow hazard early warning.

## 2. Materials and Methods

### 2.1. Study Area

The study area is mainly located in Beichuan county, with geographical coordinates of 104°23′–104°31.7′ E, 31°48.5′–31°53.5′ N, covering an area of about 140 km$^2$ (Figure 1). Since the Wenchuan earthquake in 2008, six heavy debris flow disasters have occurred, resulting in nearly 10,000 people left homeless, more than half of the buildings buried and a large area of farmland flooded [28].

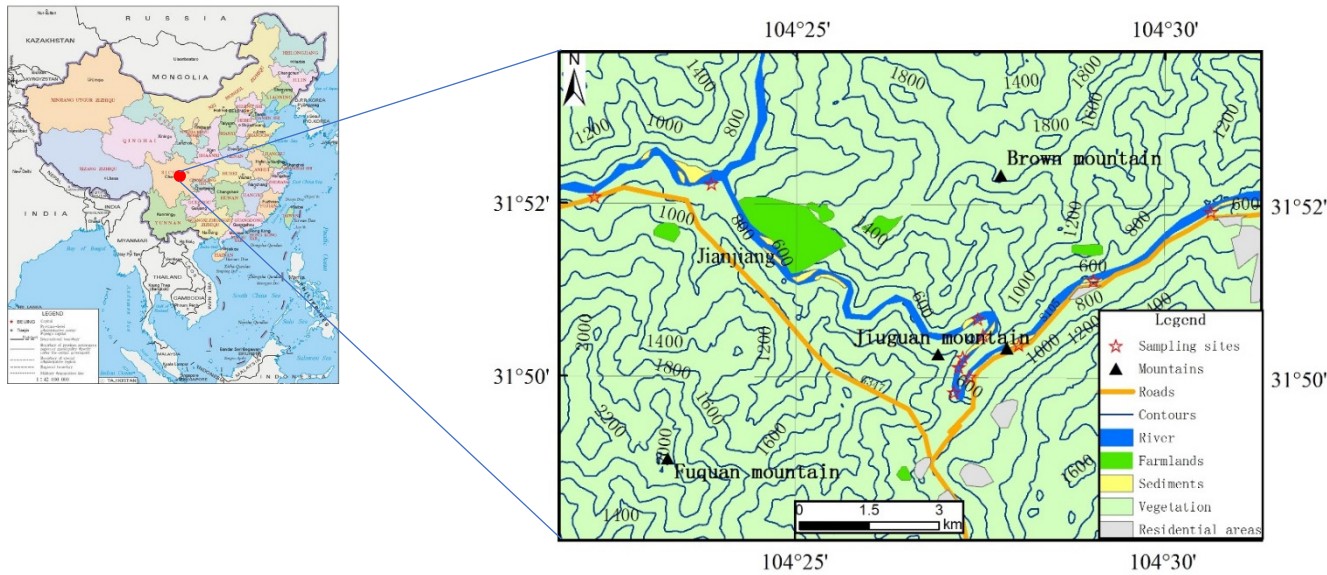

**Figure 1.** Location map of the study area.

### 2.2. Materials and Equipment

To establish a model of cohesion and its inter-controlled factors of fine-grained sediments in the study area, we collected multi-resource materials: remote sensing images, digital elevation model (DEM), soil and its parameters such as cohesion, permeability coefficient, density and particle size using the corresponding equipment listed in Table 1.

**Table 1.** Materials and equipment.

| Materials | Equipment | Manufacturer/Provider |
|---|---|---|
| Remote sensing images | Gaofen (GF) | Land satellite remote sensing application center, China |
| Digital Elevation Model (DEM) | Advanced Spaceborne Thermal Emission and Reflection Radiometer (ASTER) | Ministry of International Trade and Industry, Japan |
| Soil | Ring knife (200 mL) | Longnian Hardware Tools Store, China |
| Cohesion | ZJ strain-controlled direct shear instrument | Nanjing soil instrument factory Company Limited (Co., Ltd.), China |
| Permeability coefficient | TST-55 permeameter | Zhejiang Dadi Instrument Co., Ltd., China |
| Density | MDJ-300A solid densitometer | Shanghai Lichen Instrument Technology Co., Ltd., China |
| Moisture | Electric heating constant temperature drying oven | Shanghai-southern Electric Furnace Oven Factory, China |
| Particle size | Microtrac S3500 | American Microtrac Incorporated (Inc.) |

### 2.3. Methods

A technical flowchart of this research is shown in Figure 2, which mainly includes the steps of data acquisition, parameter measurement experiment and model establishment.

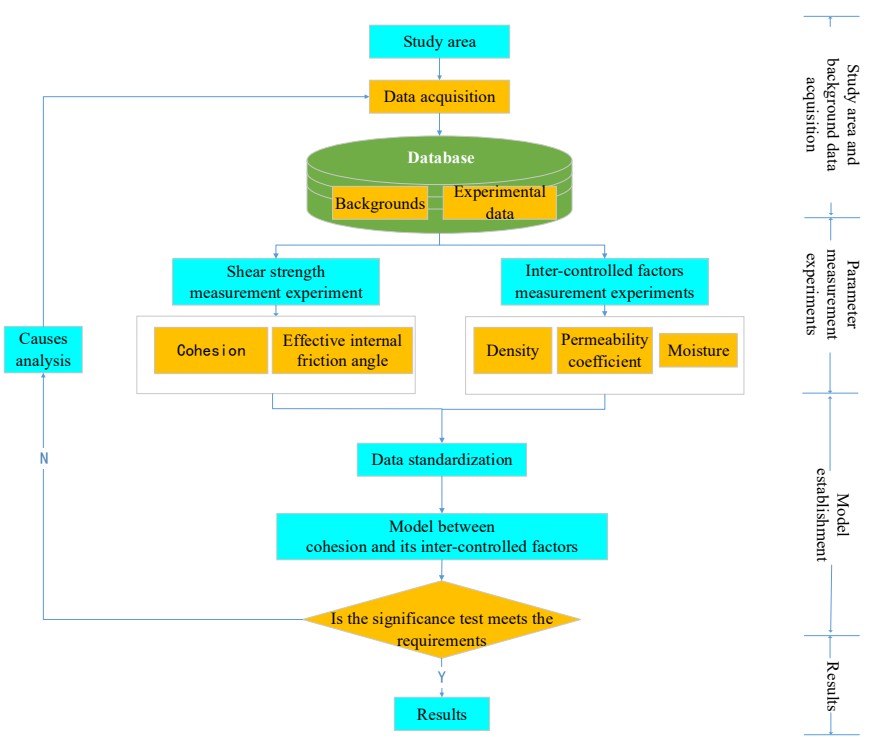

**Figure 2.** Technical flowchart.

### 2.3.1. Data Acquisition

(1) Background data

GF-2 satellite remote sensing images with a spatial resolution of 0.8 m and DEM with a spatial resolution of 30 m were acquired first. Then, a fine-grained sediment map was extracted from the GF imagery to determine the locations of sampling sites.

(2) Sample collection

From 19 to 25 March 2021, with cloudy weather and 11–15 °C temperature, 200 samples (600 mL for each sample) were collected from 11 sampling sites using ring knives according to the guide GB/T 36197-2018 [29], whose locations are shown in Figure 1.

(3) Database establishment

A database composed of remote sensing images, DEM, fine-grained sediments map, soil locations, pictures and descriptions was established according to the principles of GB/T 30319-2013 [30].

2.3.2. Parameter Measurement Experiment

Experiments were carried out according to the specification SL237-1999 [31]. Measured parameters include particle size, cohesion and its inter-controlled factors, such as effective internal friction angle, permeability coefficient, density and moisture. Detailed information about the experiments can be found in Reference [28].

(1) Particle size measurement experiment

Microtrac S3500 in Section 2.2 was used to measure particle size; the main steps include setting the sample number and parameters on the instrument, particle size automatic measurement, saving data and cleaning the pipeline.

A histogram of fine-grained sediments' particle size is shown in Figure 3. We can see that the minimum soil particle size in the study area is 0.45 um and the maximum is about 90 um, most of which is distributed in the range of 10–20 um. According to the standard for the engineering classification of soil (GB/T 50145-2007) [32], they are classified as silt loams.

Number

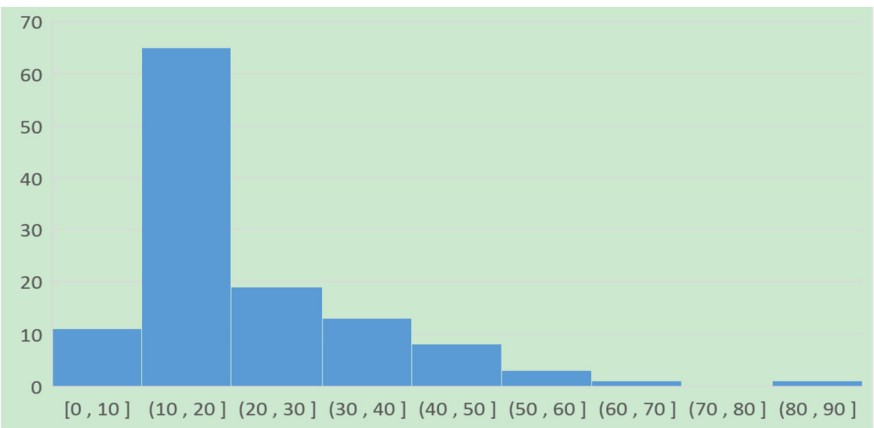

Particle size (um)

**Figure 3.** Histogram of fine-grained sediments' particle size.

(2) Cohesion measurement experiment

A ZJ strain-controlled direct shear instrument in Section 2.2 was used to measure the soil cohesion and effective internal friction angle. The main steps of the experiment include sample preparation, adding shear normal stress $\sigma$ of 50, 100, 200 and 300 kpa to obtain shear strength $\tau$, then calculating cohesion and effective internal friction angle by showing the 4 pairs of data in the coordinate system, in which $\sigma$ is on the horizontal axis and $\tau$ is on the vertical axis.

Histograms of fine-grained sediments' cohesion and effective internal friction angle are shown in Figure 4. We can see that cohesion is distributed from 13.95 to 39.55 kPa with the main range of 17.15–26.75 kPa, and the effective internal friction angle is distributed from 16.16° to 23.68°, with the main range of 18.98–21.80°.

(3) Cohesion inter-controlled factor measurement experiments

A series of measurement experiments were carried out to acquire cohesion inter-controlled factors, such as permeability coefficient, density and moisture.

Firstly, the TST-55 permeameter in Section 2.2 was used to carry out the permeability coefficient experiment, whose main steps include sample preparation, flowing water through the sample and recording related parameters such as initial water head, starting

time and the end water head. Then, we calculated the permeability coefficient according to its formula.

Number

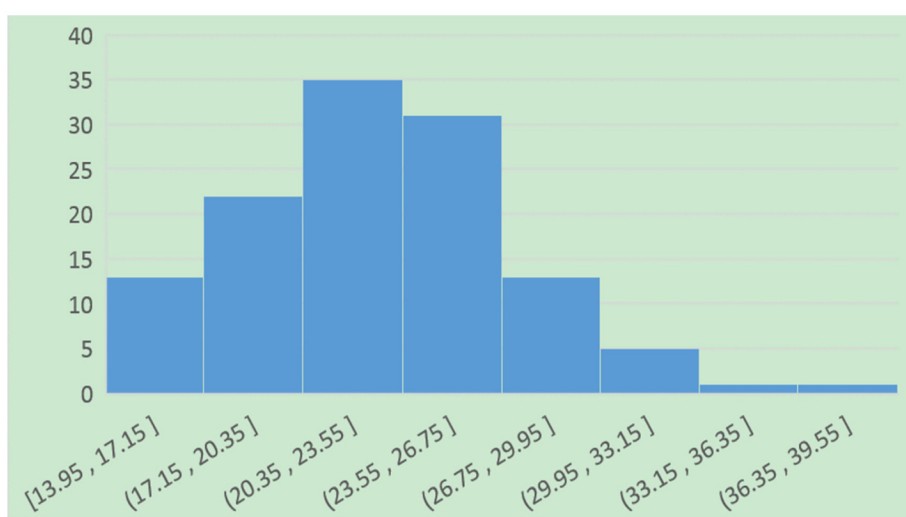

(a) Cohesion (kPa)

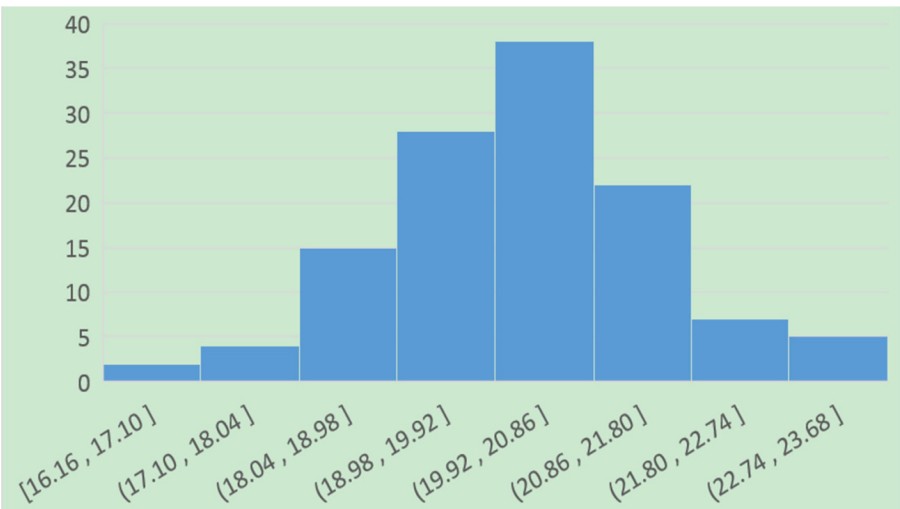

(b) Internal friction angle (°)

**Figure 4.** Histograms of fine-grained sediments' cohesion (**a**) and effective internal friction angle (**b**) in Beichuan.

A histogram of the fine-grained sediments' permeability coefficient is shown in Figure 5. We can see that the permeability coefficient is distributed from 0.47 to 2.85 m/d, with the main range of 1.15–2.17 m/d.

Secondly, the MDJ-300A solid densitometer in Section 2.2 is used to measure density; the main steps include sample preparation, weighing the sample and its bag, and then calculating density.

A histogram of fine-grained sediments' density is shown in Figure 6. We can see that density is distributed from 1.34 to 1.73 g/mL, with the main range of 1.34–1.54 g/mL.

Number

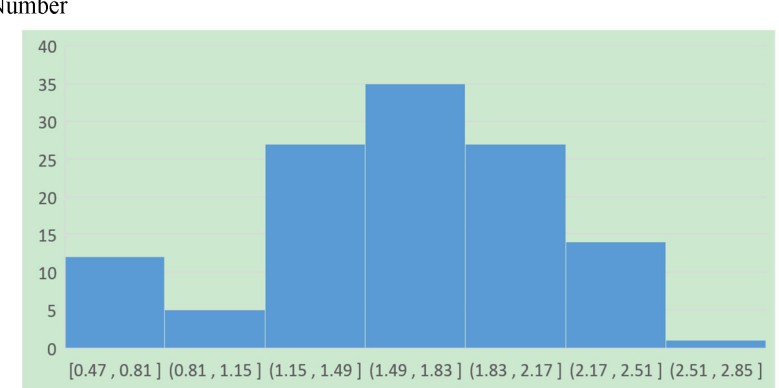

Figure 5. Histogram of fine-grained sediments' permeability coefficient in Beichuan.

Number

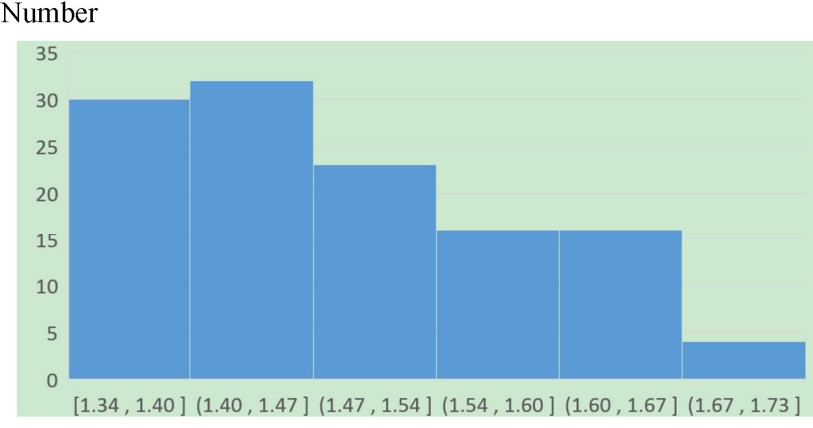

**Figure 6.** Histogram of fine-grained sediments' density in Beichuan.

Finally, an electric heating constant temperature drying oven in Section 2.2 was used to measure moisture. The main steps include sample preparation and weighting, drying the sample for more than 8 hours, weighting dried samples and then calculating moisture.

A histogram of fine-grained sediments' moisture is shown in Figure 7. We can see that moisture is distributed from 4.01% to 30.69%, with the main range of 4.01–12.02%.

Number

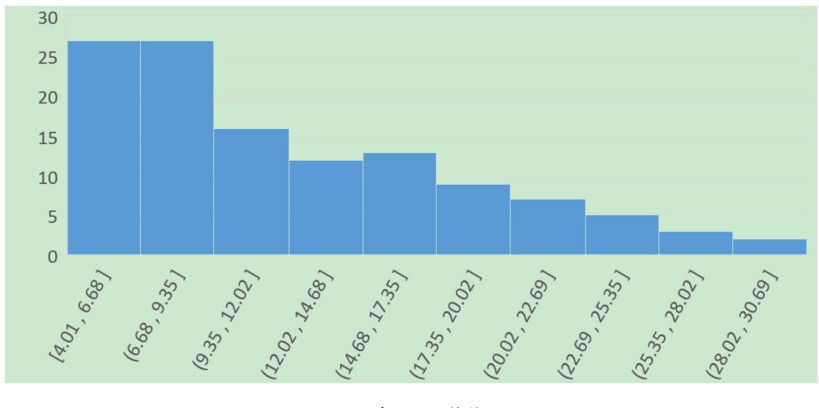

**Figure 7.** Histogram of fine-grained sediments' moisture in Beichuan.

### 2.3.3. Model

(1) Data standardization

To eliminate the impacts of magnitude dimensions and orders for different units, data standardization was carried out using Equation (1).

$$z_{ij} = (x_{ij} - \overline{x_i})/s_i \tag{1}$$

where $z_{ij}$ is the standardized value, $x_{ij}$ is the measured value, $\overline{x_i}$ is the mean value and $s_i$ is the standard deviation.

(2) Close factors to cohesion selection

As shown in Table 2, in order to determine the factors close to cohesion, the correlation coefficients between cohesion and effective internal friction angle, permeability coefficient, density and moisture were calculated.

**Table 2.** Coefficients between cohesion correlated to its factors.

|  | **Effective Internal Friction Angle (°)** | **ln(p) (m/d)** | **Density (g/cm³)** | **Moisture (%)** |
|---|---|---|---|---|
| Cohesion (KPa) | −0.66 | −0.58 | 0.36 | 0.32 |

p: permeability coefficient; ln: natural logarithm.

From Table 2, we can see that with the correlation coefficients of −0.66, −0.58, 0.36 and 0.32, the effective internal friction angle, logarithm of permeability coefficient, density and moisture, respectively, are related to cohesion. The correlation coefficients of the former two are negative, indicating that cohesion decreases with the increase in effective internal friction angle and permeability coefficient, while the others are positive, indicating that cohesion increases with the increase in density and moisture.

Figure 8 shows the fitting relationship between cohesion and its inter-controlled factors. From which, we can see that the fitting determination coefficient ($R^2$) between cohesion and the permeability coefficient is 0.31, which is less than 0.34 of the fitting determination coefficient between cohesion and the logarithm of permeability coefficient, logarithmic transformation on the permeability coefficient should be made before regression. We also applied logarithms on the effective internal friction angle and density, and then calculated determination coefficients between cohesion with them. The results showed that their logarithmic determination coefficients were 0.44 and 0.12, respectively, which are no more than those of 0.44 and 0.13 in their linear regression format.

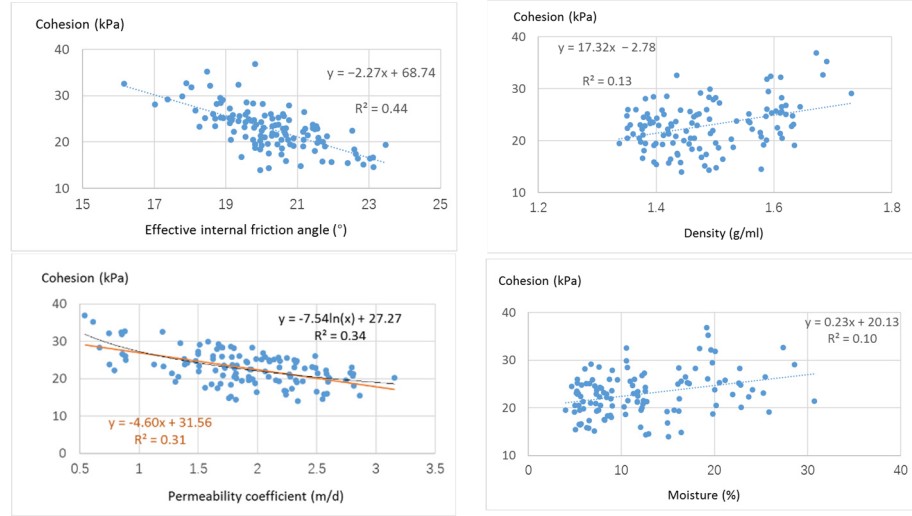

**Figure 8.** Fitting relationship between cohesion and its inter-controlled factors.

According to the principles of the statistical significance test, when the correlation coefficient (R) is more than 0.3 and the *p* value of the *t*-test is no more than 0.05, the factor passes the *t*-test. However, the t-test on the correlation between cohesion and its inter-controlled factors (Table 3) showed that the *p* value of moisture was 0.45, which is much more than 0.05, indicating that moisture did not pass the *t*-test, and it was then deleted from the cohesion inter-controlled factors.

**Table 3.** The *t*-test of the correlation between cohesion and its inter-controlled factors.

| | Coefficient | *p* Value | Lower Limit 95.0% | Upper Limit 95.0% |
|---|---|---|---|---|
| Intercept | 22.91 | 0.00 | 22.40 | 23.43 |
| Density | 0.86 | 0.05 | −0.03 | 1.76 |
| ln(p) | −1.59 | 0.00 | −2.23 | −0.96 |
| Effective internal friction angle | −2.49 | 0.00 | −3.04 | −1.95 |
| Moisture | −0.33 | 0.45 | −1.20 | 0.54 |

p: permeability coefficient; ln: natural logarithm.

Therefore, close factors to cohesion are effective internal friction angle, logarithm of permeability coefficient and density.

(3) Model establishment

After selecting the close factors to cohesion, a model between them is established using the least-squares multivariate statistical method.

Firstly, a model between cohesion and each inter-controlled factor is established by the correlation fitting method, such as scatter plot analysis in Microsoft Excel. Then, a transformation on each factor is carried out according to the model between cohesion and each inter-controlled factor so that the cohesion and each inter-controlled factor are linearly correlated. Finally, a model of cohesion and its inter-controlled factors is established by the least-squares multivariate statistical method, whose principles are as follows.

By minimizing the summation of squared errors, the least-squares multivariate statistical method finds the best matching function.

$$y = f(x, w)$$

In order to determine $w$, the function can be solved as

$$L(y, f(x, w)) = \sum_{i=1}^{n} |y_i - f(x_i, w_i)|^2 \tag{2}$$

where $w_i$ ($i = 1, 2, \ldots, n$) can be calculated by minimizing the function.

## 3. Results and Discussion

### 3.1. Results

Using the experimental data, the model of cohesion and standardized inter-controlled factors is established as

$$y = 22.91 + 0.62x_1 - 1.57x_2 - 2.48x_3$$

where $y$ is cohesion, $x_1$ is the normalized density, $x_2$ is the normalized logarithm of permeability coefficient ln(p) (p is the permeability coefficient) and $x_3$ is the normalized effective internal friction angle.

The fitting correlation of the model is shown in Figure 9.

From Figure 9, we can see that the model's determination coefficient ($R^2$) is 0.61, indicating that the corresponding correlation coefficient (R) is 0.78. In the model, the absolute value of each variable coefficient indicates its closeness to cohesion. Because the absolute coefficients of normalized density, normalized logarithm of permeability coefficient and

normalized effective internal friction angle are 0.62, 1.57 and 2.48, respectively, the closest parameter to cohesion is the effective internal friction angle, followed by the logarithm of permeability coefficient and density in the study area.

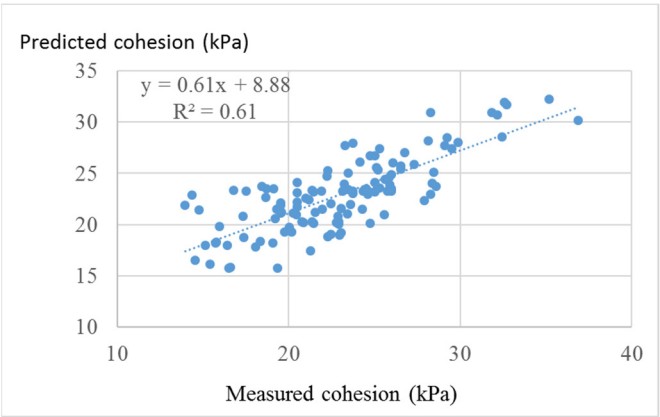

**Figure 9.** Fitting correlation between the measured and the predicted cohesion.

The results of the *t*-tests on each regression coefficient are shown in Table 4. We can see that except for the *p* value of density, which is 0.05, others are less than 0.01, indicating that cohesion has a significant correlation with its inter-controlled factors.

**Table 4.** The *t*-tests on the model.

| | Coefficient | *p* Value | Lower Limit 95.0% | Upper Limit 95.0 |
|---|---|---|---|---|
| Intercept | 22.91 | 0.00 | 22.40 | 23.43 |
| Density | 0.62 | 0.05 | 0.01 | 1.22 |
| ln(p) | −1.57 | 0.00 | −2.20 | −0.93 |
| Effective internal friction angle | −2.48 | 0.00 | −3.03 | −1.93 |

p: permeability coefficient; ln: natural logarithm.

### *3.2. Discussion*

Cohesion is negatively correlated to effective internal friction angle and logarithm of permeability coefficient and positively correlated to density; the reasons for this are analyzed as follows.

(1) The fixed shear strength of a sample makes a negative correlation between cohesion and the effective internal friction angle.

The effective internal friction angle is determined by the friction resistance and linkage between soil particles. Larger factors that lead to the increase in the soil friction angle—such as a coarser surface, more edges and corners and greater spaces between soil particles—result in the weakening of the gravity between soil particles and the reduction in cohesion [23,33].

In a sample, the correlation of cohesion and the effective internal friction angle can be expressed by

$$\tau_f = \sigma tg\varphi + c \tag{3}$$

where $c$ is the cohesion (kPa), $\tau_f$ is the shear strength (kPa), $\phi$ is the effective internal friction angle (°), $\sigma$ is the normal pressure (kPa) and $\sigma tg\phi$ is internal friction.

Therefore, when shear strength $\tau_f$ and normal pressure $\sigma$ are fixed, the greater the effective internal friction angle is and the smaller the cohesion becomes, thus leading to a negative correlation between them.

(2) Attraction controlled by distance between soil particles makes a negative correlation between cohesion and the permeability coefficient, and a positive correlation between cohesion and density.

The value of the permeability coefficient mainly reflects the number, size and connectivity of soil pores. The greater the permeability coefficient is, the larger the distance between soil particles becomes, leading to smaller attraction and the weakening of cohesion, and thus leading to a negative correlation between cohesion and the permeability coefficient [23].

The greater the soil density is, the smaller the distance between soil particles is and the greater the mutual attraction between particles becomes, leading to greater soil cohesion. Therefore, there is a positive correlation between cohesion and density [34].

(3) The correlation between cohesion and moisture is not significant in the study area.

Many studies showed that soil moisture has some influence on cohesion, but their correlation varies with the content of water and soil components: (1) with the increase in water content, soil cohesion increases first and then decreases [35], and (2) soil cohesion differs with the soil components. For example, it is higher in red soil, mid-range in kaolin and the lowest in sandy soil [36].

Therefore, the correlation between moisture and cohesion is complex, as cohesion does not show a consistent trend with the change in moisture, thus leading to a small correlation coefficient between them. Furthermore, the soil type in the study area is silt loam, in which the correlation between cohesion and moisture is not as significant as in red soil or kaolin in other places.

## 4. Conclusions

We designed a series of soil experiments to establish a model of cohesion and its inter-controlled factors. The main conclusions are as follows.

(1) The cohesion inter-controlled factors of fine-grained sediments in Beichuan debris flow were discovered.

The selection of close factors to cohesion showed that with the correlation coefficients of $-0.66$, $-0.58$ and $0.36$, effective internal friction angle, logarithm of permeability coefficient and density, respectively, were related to cohesion. Therefore, the cohesion inter-controlled factors of fine-grained sediments in Beichuan include effective internal friction angle, logarithm of permeability coefficient and density, as analyzed in Section 3.2.

(2) A model of cohesion and its inter-controlled factors in Beichuan debris flow was established.

A model of cohesion and its inter-controlled factors (effective internal friction angle, logarithm of permeability coefficient and density) in Beichuan debris flow was established by the least-squares multivariate statistical method. The results show that the absolute coefficients of normalized density, normalized logarithm of permeability coefficient and normalized effective internal friction angle were $0.62$, $1.57$ and $2.48$, respectively, indicating that the closest parameter to cohesion is the effective internal friction angle, followed by the logarithm of permeability coefficient and density in the study area. The results of $t$-tests on each regression coefficient showed that except for the $p$ value of density, which was $0.05$, those of other factors were less than $0.01$, indicating that cohesion had a significant correlation with its inter-controlled factors.

(3) The quantitative model of cohesion and its inter-controlled factors provides a scientific basis for debris flow hazard early warning.

Fine sediments with particle sizes less than 2 mm are easily transported by water, especially during high-intensity rainfall or rapid snow melt. Thus, these materials play an important role in the debris flow early warning system. Debris flow early warning needs to quickly detect the stability of these fine-grained sediments, being one of the factors controlling disaster scales.

Cohesion reflects the strain capacity of soil to resist external stress, and is closely related to soil stability. Although cohesion varies with water content, we can quickly obtain its value by the quantitative model presented in this article when the fine-grained sediments on the surface are in their natural state. Then, dangerous areas can be identified by the early warning system according to the value of soil stability estimated by cohesion.

In other words, less stability indicates higher danger, and thus provides a scientific basis for debris flow early warning.

**Author Contributions:** Conceptualization, Q.W. and J.X.; methodology, Q.W., J.X., J.Y. and P.L.; validation, Q.W., J.X. and P.L.; formal analysis, Q.W. and J.X.; investigation, Q.W., J.X., P.L., D.C. and W.X.; data curation, J.X., J.Y. and P.L.; writing—original draft preparation, Q.W., J.X. and J.Y.; writing—review and editing, Q.W.; visualization, D.C.; supervision, W.X.; project administration, Q.W.; funding acquisition, Q.W. All authors have read and agreed to the published version of the manuscript.

**Funding:** This research was funded in part by the National Natural Science Foundation of China (grant number 42071312), the Innovative Research Program of the International Research Center of Big Data for Sustainable Development Goals (grant number CBAS2022IRP03), the Hainan Hundred Special Project (grant number 31, JTT [2018]), the National Key R&D Program (grant number 2021YFB3900503), the Special Project of Strategic Leading Science and Technology of the Chinese Academy of Sciences (grant number XDA19090139), the Second Tibetan Plateau Scientific Expedition and Research (STEP) (grant number 2019QZKK0806) and the Hainan Provincial Department of Science and Technology (grant number ZDKJ2019006).

**Institutional Review Board Statement:** Not applicable.

**Informed Consent Statement:** Not applicable.

**Data Availability Statement:** Not applicable.

**Acknowledgments:** We express our acknowledgments to Chuanxin Li and Shuo Gao in China University of Geosciences (Beijing) for their hardware supporting. We also express our acknowledgments to reviewers and editors for their good comments and suggestions to make the article better. Finally, we express our acknowledgments to Yu Chen, Bihong Fu and Pilong Shi in the Aerospace Information Research Institute, Chinese Academy of Sciences for their hardware supporting.

**Conflicts of Interest:** The authors declare that they have no conflict of interest.

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
