# Peer review of "A Model between Cohesion and Its Inter-Controlled Factors of Fine-Grained Sediments in Beichuan Debris Flow, Sichuan Province, China"

_sustainability, doi:10.3390/su141912832_

Round 1
Reviewer 1 Report (New Reviewer)
The paper discussed an interesting topic in debris flow. As physically based models are mostly time consuming and requires a large set of input parameters, which are not always available, some studies tried to simplify the calculation by empirical approach.
However, the paper has some flaws in writing style, structure, and presents too many unclear information. Major revision is needed, and a recommendation to accept this paper can only be given once all issues below are addressed.
Major Comments
(1) The paper needs to be checked by professional English proofread service. Although they mostly make sense, I found it hard to understand them by one time reading.
(2) Abstract is unclear, please check the detailed comment section for revision.
(3) In Section 2.2 and 2.3, there were so many acronyms included, and the authors did not bother to explain it at all. It is confusing as the readers aren’t familiar. Please explain more about the acronym and gives detail information about the data as follows.
Please explain in detail about high remote sensing data: what is that? What is the resolution, who are the data provider. What are the geological data, what are the meteorological data, what do you mean by “other background data”.
What is ResNeXt? What is FPN?
they are all unclear and hardly read.
(4) While the first step in methods (Data acquisition) is based on previous research, I suggest the authors to help the readers to understand it by just reading this paper. Hence, and an appendix summarizing the method should be added to the manuscript.
This is important because the manuscript is planned to be published under “Applications of GIS and Remote Sensing in Soil Environment Monitoring SI”. In my opinion, only this step is GIS and remote sensing related, the others are mainly empirical.
(5) The Article needs to be restructured as follows
1. Introduction
2. Materials and Methods
2.1 Study area
2.2 Materials and equipment
2.3 Methods
1 Methods should cover all the steps include in Figure 2.
2.3.1 Data acquisition
2.3.2 Parameter measurement experiment
And so on
Current structure is so confusing. what is the difference between 2.2 and 2.3. They look repetitive. Example:
Page 3 lines xx, Data acquisition,
Page 4 lines xx, 2.3 Data acquisition and processing
Aren’t they the same?
there are also point (2) in page 3 lines 83 and point (3) in page 4, lines 116. Both seem like the same step: Parameter measurement experiments
Please distinguished results from discussion
3 Results and Discussion
3.1 Results
3.2 Discussion
(6) At the current state, there seems very little discussion in section 3. Authors must elaborate more, implying
(1) How can their empiric model is explained by physics and mechanisms of debris flow
(2) how their results can contribute to debris flow studies (related to point7 below).
As well as most of the information in conclusion (lines 228-243) should belong to result and discussion section.
(7) The authors took 200 samples, but none of the results of soil properties they write in section 2.2 and 2.3 were presented in the results. I think histograms to describe the distribution of data is necessary. Even if the authors decided to put them in appendix or supplementary materials.
(8) As the authors explain in the abstract that the quantitative model between cohesion and its inter-controlled factors can significantly contribute to debris flow hazard early warning, authors must explain (in the discussion) about how their model can be used for early warning of debris flow.
(9) Although this study is about developing an empirical model of cohesion, the authors should also discuss the theory, physics and mechanism of debris flow (related to my previous comment (6) )
Some details comments are given below:
1. Line 16-19.
Comment: I am not so clear with the passage. It is written that cohesion was the main inter-controlled factor of fine-grained sediments stability, but the next lines mention about quantitative model between cohesion and its inter-controlled factors. What does “its” in lines 18-19 refers to? fine-grained sediments or cohesion? What are these factors? Can the authors elaborate it?
I suggest the authors rearrange first 4 lines in abstract by explaining the inter-controlled factors for clarity.
2. Lines 22-25,
the authors should declare the parameterization of the least square multi-variate.
Please consider rearrange the sentence for clarity, example:
We used parameter a, parameter b, parameter c, etc to establish a quantitative model between cohesion and its inter-controlled factors. Fitting of the experimental cohesion and the quantitative models showed that the determination coefficient was 0.6125
And why did the authors use four digits after decimal? In lines 28-29, all numeric results are presented in three- and two-digits decimals. Please make them uniform.
3. Line 34: Incorrect. Debris flow is not floods. It is a sediment flow that is more relevant to landslide than flood. In hydraulic engineering, most debris flow models are based non-newtonian fluid assumptions, while flood is considered as Newtonian fluid (treated as clear water). Please refer to Takahashi (2014)
4. Lines 41: …. particle size less than 2 mm.
Comments:
What do you mean by size? Please clarify if it is diameter. And please explain how the fine-grained sediment is the starter. Isn’t debris flow characterized by heterogenous GSD. Some deadly debris flows could consists of boulders and larger .
5. Lines 46-47: intensity, runoff, etc., and topographic conditions, such as slope, surface coverage, structure etc.;.
Comment:
Please elaborate what is etc. in this lines and any other lines when it occurs.
6. Lines 53-54.
Comment:
Please also define effective internal friction angle, permeability coefficient, density and moisture and how they will affect cohesion.
7. Lines 63-65: Since Wenchuan earthquake took place in 2008, six heavy debris flow disasters have occurred, resulting in nearly 10,000 people left home, more than half of the buildings buried, and a large number of farmland flooded.
Comment:
What is the source of this important information?
8. Figure 1.
Comments:
(1) Is this the entire map Beichuan county? Please add another map that can help reader easily locate Beichuan County or study area from Chinese region. Can be an insert or a map where Beichuan is the zoom area.
(2) As debris flow is a type of gravity sediment flow and most models include elevation/height please add a figure of the elevation (DEM) of the study area or at least the vertical profile.
9. Line 108. When the accuracy is greater than 85% …
Comments:
What does the accuracy in this sentence refer too? is it coefficient determination or correlation? and why should it be grater than 85%?
10. Line 113. From March 19 to 25, 2021, with a volume of 600ml for each sample, 200 samples were collected from 11 sampling sites using ring knives, whose locations are shown in Figure 1.
Comment:
Please describe the weather/climate during the sampling period.
11. Lines 127-128 and Lines 141-143
With a scope of 13.95-39.55kPa in cohesion and 16.16-23.68 in effective internal friction angle, soil in the study area belongs to the silt loams [29].
The results showed that with a scope of 0.47-2.85m/d in permeability coefficient, 1.34- 1.73 g/ml in density and 4.01-30.69% in moisture, soil in the study area belongs to the silt loams.
Comments:
(1) Can you explain how to determine soil types (texture) from cohesion, effective internal friction angle, permeability coefficient, density, and moisture
(2) Though all sampling points seems to be located near the stream, why the gap of the soil moisture is so significant?
12. Line 151 … using the least-squared multi-variate statistical
Comment: Why the authors select this method, why didn’t you use another method, such as polynomial regression or else?
13. Lines 156-160 Table 1.
Comment:
Please explain or describe the information in this table 1 in the main text. What is the interpretation of negative and positive correlation?
14. Lines 163-172 Figure 3 and the paragraph below.
Comment:
Why Normal Logarithmic was only applied to permeability coefficient and not for the other inter-controlled factors?
15. Line 168: Then, moisture is eliminated out for it does not pass the t-test
Comments:
(1) There is not any single sentence in the method that said all parameters must pass t-test before including them in the model.
I suggest adding the results of t-test for all parameters in the appendices. I don’t think they are the same with Table 2, because soil moisture is already excluded here.
(2) Most widely assumption is debris flow behavior are strongly determined by pore fluid pressure and granular temperature (Iverson, 1997, Iverson and LaHusen, 1997, and many more). How can the authors justify the exclusion of soil moisture as the insignificant parameter in the model, theoretically (not statistically).
16. Lines 187-188: From Figure 4, we can see that the model’s determination coefficient (R2) is 0.6125 indicating the corresponding correlation coefficient is 0.78.
Comment:
this is related to my previous comment. Why is the R2 = 0.6125 and correlation= 0.78 is considered proving the model effectivity. Did not the threshold was 0.85?
17. Lines 200-203: The larger the factors that lead to the increase of the soil friction angle, such as the coarser the surface, the more edges and corners become, the greater the space between soil particles are, which resulted in the weakening of the gravity between soil particles and the reduction of cohesion[20].
Comment:
What does it imply to debris flow? Please elaborate, adding some more references.
18. Line 252:
..the model is effective in discovering the mechanism of soil cohesion …
Comments: The authors never explain the mechanism.
Minor Comments:
(1) Only equation (1) is numbered. Please revise for the other equations.

Author Response
Please see the attachment.

Reviewer 2 Report (New Reviewer)
The authors studied the cohesion inter-controlled factors of fine-grained sediments in Beichuan debris flow and proposed a model to predict cohesion. This study is very interesting and worthy publishing. My comments are listed as follows:
1. The introduction part is so simple. I do agree that it is important to develop debris flow hazard early warning system. But I suggest the authors to add a part to summarize latest progress for different numerical simulations method, which is essential for development of hazard early warning system. The following papers are listed for authors:
SPH-based simulation of flow process of a landslide at Hongao landfill in China. Natural Hazards, 93(3), 1113-1126.
Simulation of flow slides in municipal solid waste dumps using a modified MPS method. Natural hazards, 74(2), 491-508.
Coupled moving particle simulation–finite-element method analysis of fluid–structure interaction in geodisasters. International Journal of Geomechanics, 21(6), 04021081.
Liquid-gas-like phase transition in sand flow under microgravity. Microgravity Science and Technology, 27(3), 155-170.
2. In figure 3, there are figures presented permeability coefficient. If you plan to compare different fitting lines, just plot different line in one figure is enough.
3. I cannot understand why there is a relationship between effective internal friction angle and cohesion. Could the authors give us much more details?
Round 2
Reviewer 1 Report (New Reviewer)
Dear Authors,
I would like to express my appreciation for all work you did to address each of my comments. After reading the revised manuscript I would like to suggest some minor changes and a major comment in order to improve it. Thank you and good luck.
Sincerely,
Major Comment:
Lines 42-44.
Comment:
It is good that the authors added previous studies in numerical debris flow models. However, the authors need to describe what are the weaknesses of those models or what is still lacking in those studies. Authors must also emphasize how can their study contribute to filling the gaps or (perhaps) enhancing the debris flow models.
Minor Comments:
1. Lines 61: permeability coefficient(the unit flow …
Comment: Please add space after coefficient
2. Line 63: density(mass per unit
Comment: Please add space after density
3. Lines 84: Digital Elevation Model(DEM)
Comment: Please add space after Model
4. Lines 93-94: … and Digital Elevation Model (DEM)
Comment: just write down DEM here, as it has been mentioned earlier
5. Line 257 equation (3)
Comment: I think the line indent here is not correct.
6. Line 258
Comment: Please type parameter c or cohesion in Italic format.
7. Figure 8, line 185.
Comment: Please check the regression equations. Digits after decimals are not uniform.
8. Another Comment: Just a writing style, the author can opt out for not following this suggestion. The way the authors lay Figures 3-7 near the explanation is good, so readers can directly read and interpret the histograms. But for efficiency and a better layout, perhaps Figures 3-7 can be put in one bigger chart (multi-panel style).

Author Response
Please see the attachment.

Reviewer 2 Report (New Reviewer)
The paper can be published as it is.
Author Response
1. The paper can be published as it is.
Response 1: Thanks for your good comments and suggestions to make the article sustainability-1891222 better!
This manuscript is a resubmission of an earlier submission. The following is a list of the peer review reports and author responses from that submission.
Round 1
Reviewer 1 Report
Dear authors, the topic of the manuscript is very interesting and worthy of researching. However, in my opinion it is necessary to provide additional information and clarify some aspects in order to be accepted for publication. In the following list are some general suggestions that may be considered by the authors:
1. Authors should include more theoretical information about the connection made between the various parameters they used.
2. Authors should provide more information about the least-squares multivariate statistical analysis they used.
3. Figures should be improved. Provide a better location and sampling maps.
4. Authors should analyze and compare their results with research of others.
5. The authors report.. After the model between cohesion and its inter-controlled factors is established, whether to improve it or not is determined according to its accuracy. What is that level of accuracy and be which way the model is improved? The authors should provide more information concerning the technical flow.
6. Authors should include more information concerning the linkage of cohesion and flow debris early systems.
Hope my comments and suggestions are useful.
Best regards.
Reviewer 2 Report
This research established an empirical formula between the cohesion of fine-grained sediments in Beichuan debris flow gullies and other parameters such as effective internal friction angle, density, and permeability coefficient based on a series of experiment data. In my opinion, this manuscript cannot be accepted for publication because it is presented in a very rough way. Some specific comments are as follows:
1) The authors state that this research can play an important role in the debris flow hazard early warning, but they don’t explain how to predict debris flow disasters based on the empirical formula.
2) Lines 87 and 88, “The improved depth learning method was used to extract the map of fine-grained sediments in the study area.” But no explain about the depth learning method was provided.
3) Lines 98 and 99, “a model between cohesion and each inter-controlled factor is established by a statistical method”. What kind of statistical method was used?
4) Lines 111-112, “Geological maps, meteorological and hydrological maps, DEM/topographic maps, traffic maps, etc., were collected to provide backgrounds for this research”, but they don’t provide those maps.
5) The data used in this research were obtained through a series of geotechnical experiments, but the authors don’t describe the experiments.
6) Figures in this manuscript are in very poor quality, such as Fig.4 and 5.
7) Line 177, what are the physical meaning of the coefficients in this equation? Is this equation applicable for the soil in other position?